# Education in Radiation Oncology—Current Challenges and Difficulties

**DOI:** 10.3390/ijerph19073772

**Published:** 2022-03-22

**Authors:** Camil Ciprian Mireștean, Roxana Irina Iancu, Dragoș Petru Teodor Iancu

**Affiliations:** 1Department of Medical Oncology and Radiotherapy, University of Medicine and Pharmacy of Craiova, 200349 Craiova, Romania; mc3313@yahoo.com; 2Department of Surgery, Railways Clinical Hospital, 700506 Iasi, Romania; 3Oral Pathology Department, “Gr. T. Popa” University of Medicine and Pharmacy, 700115 Iasi, Romania; 4Department of Clinical Laboratory, St. Spiridon Emergency Hospital, 700111 Iasi, Romania; 5Department of Medical Oncology and Radiotherapy, “Gr. T. Popa” University of Medicine and Pharmacy, 700115 Iasi, Romania; dt_iancu@yahoo.com; 6Department of Radiation Oncology, Regional Institute of Oncology, 700483 Iasi, Romania

**Keywords:** radiation oncology, radiotherapy, education, students, curriculum, mentorship, training

## Abstract

The evolution and development of radiotherapy in the last two decades has meant that postgraduate medical training has not kept up with this rapid progress both in terms of multidisciplinary clinical approaches and especially in terms of technological advances. Education in radiation oncology is a major priority in the context of the rapid development of radiotherapy, including advanced knowledge of radiobiology, radiation physics and clinical oncology, anatomy, tumor biology and advanced medical imaging. In this context, the lack of training in radiation oncology in the curricula of medical faculties may have detrimental consequences for the training of residents in radiotherapy but also in their choice of specialty after completing their university studies. There is a clear gap between resident physicians’ actual and required knowledge of radiotherapy, and this requires urgent remediation. In the context of technical advances in imaging-guided radiotherapy (IGRT) and new radiobiology data, a balanced approach divided equally between general oncology, clinical radiation oncology, radiation oncology technology, medical physics and radiobiology, anatomy and multimodal imaging, including mentorship could bring educational and career choice benefits for students of radiation oncology.

## 1. Introduction

The evolution and development of radiotherapy in the last two decades has meant that postgraduate medical training has not kept up with this rapid progress both in terms of multidisciplinary clinical approaches and especially in terms of technological advances. These advances include the use of computer tomography (CT) simulators in radiotherapy treatment planning, the concept of image-guided radiotherapy (IGRT) and modern irradiation techniques, among which we mention 3D-conformal (3D-CRT) and more recently Intensity Modulated Radiation Radiotherapy (IMRT) or Intensity Modulated Volumetric Arc Therapy (VMAT). Additionally, techniques based on a high geometrical conformity of radiation doses on the target volume, modern brachytherapy using three-dimensional reconstruction of structures, stereotactic brain and body radiosurgery techniques, and advances in modeling and understanding of radiobiology are challenges to which radiotherapy education must respond urgently. In order to understand the major difficulties in the training and education of a clinician specialized in radiation oncology, we will briefly mention some novel elements essential to understanding the principles of radiation treatments in oncological diseases [1,2,3].

## 2. New Challenges and Trends with Impact on Radiation Oncology Education

Seen as a multidisciplinary approach, the introduction of molecular targeted therapies, and more recently of immunotherapy in sequential or concomitant association with traditional chemotherapy, brings a new spectrum of toxicity and dramatic changes in prognosis. These revolutionary therapies have dramatically changed the course of the disease, with long-term survival even in the recurrent or metastatic stage. The increase in life expectancy of patients receiving multimodal treatments including radiation therapy amplifies the risk of severe side effects, which may compromise quality of life (QoL) or even be life-threatening. In countries such as the United Kingdom, the specialty called “clinical oncology” with an educational program of 5 years includes the non-surgical therapeutic approach to cancer (radiotherapy and systemic oncological therapy), whereas in countries where “radiation oncology” is a distinct specialization, the training includes only one module of medical oncology.

The new concept of guided imaging radiotherapy (IGRT) requires advanced knowledge of medical imaging. This requires not only knowledge of anatomy and an understanding of radiographs, but also an advanced understanding of CT imaging and magnetic resonance imaging (MRI). Recently, the concept of “biological dose painting” has brought hybrid medical imaging not only to the diagnosis of cancer but also to radiation therapy treatment planning. Functional MRI and positron emission tomography (PET) are included in the radiotherapy treatment plan to refine the irradiation dose according to metabolism, hypoxia, oxygen and blood diffusion, and other metabolic parameters of tumors.

The use of standard therapeutic techniques with a high level of target volume conformity requires a thorough understanding of the radiation ballistics associated not only with high conformation target volumes of the radiation dose but also with the risk of “geographic miss”. From the point of view of radiobiology and medical physics, challenges arise with the new modulators of both intrinsic and extrinsic radio-sensitivity so that the genetic and molecular peculiarities of each tumor and modern therapies influence the tumor response to irradiation. In this context, the development of mathematical radiobiological models to better characterize the response of the tumor and healthy tissue to treatment becomes a priority even if the linear-quadratic model (LQ) will remain the cornerstone of radiobiology. In addition, the use of altered fractionation schemes and the implementation of hypo-fractionation and stereotactic radiosurgery require intensive use of LQ-based equivalence formulas. Appreciation of the acute and late effects of irradiation becomes a necessity in the context of using altered fractionation schemes and complex radiobiological models, such as tumor control probability (TCP) and normal tissue complication probability (NTCP), to accurately estimate the tumor response probabilities to irradiation and toxicities to radiosensitive organs at risk (OARs) [2,4,5,6,7].

The adaptation of the LQ model for different clinical situations is masterfully exemplified by Jones et al. the authors proposing a radio-sensitivity modifying “x” correction factor. The increased toxic effects of irradiation can be quantified in various clinical scenarios, such as the use of chemotherapy cyclophosphamide, methotrexate, and fluorouracil (CMF) in breast cancer, as an aggravating factor for subcutaneous fibrosis. Fibrosis of the shoulder is another toxicity that justifies the use of the correction factor, which is added to the equivalent dose in 2 Gy for women over 60 years treated with axillary nodes irradiation. In these two clinical situations, the correction factor x has the values 1063 and 1033, respectively. The x-value is much higher for the additional gastrointestinal toxicity resulting from abdominal surgery in lymphoma patients (x = 1.18). Another method to assess the radiobiological impact of different clinical settings is to assimilate the toxic effect of each clinical and therapeutic feature with an additional value of toxic (alpha/beta ratio = 3), biologically equivalent dose (BED). The relevant values for the clinical hypotheses mentioned above are 6.48Gy, 3.61Gy and 17.73Gy, respectively. These examples illustrate the need for a deep understanding of radiobiology in the education process of the radiation oncologist clinician [8].

Although redoubtable toxicities, such as radiation myelitis, are rare in today’s era of conventional radiation therapy due to the implementation of image guided radiotherapy (IGRT) and the limitation of radiation doses received by radiosensitive structures through dosimetric constraints, advances in multimodal treatment including chemotherapy, immunotherapy and targeted molecular therapy are potential factors for a synergistic increase in toxicity. This can be a relevant fact in the context of a general improvement of the cancer patients’ prognosis. Even in palliative settings, the combination of radiation therapy with new target therapies can increase not only the therapeutic benefit but also the toxicity. An example is the association of braf inhibitors with the irradiation of brain metastases in malignant melanoma. The concomitant treatment includes the risk of severe skin toxicity on the scalp. The risk of life-threatening aspiration pneumonia associated with severe dysphagia, but also amplificated by collateral toxicities such as xerostomia is a topic of interest, especially for young patients with oro-pharyngeal cancer associated with an HPV infection who have the potential for long-term survival and require a better quality of life. Rectal bleedings after dose-escalated treatment of prostate cancer by external beam radiotherapy and intestinal brachytherapy, the risk of late cardiac mortality for young patients treated with anthracyclines or anti-HER2 therapy, and radiotherapy for early-stage breast cancer are just a few situations that justify the need for an in-depth understanding of all aspects related to radiotherapy and radiobiology of tumors and healthy tissues. They illustrate the need for advanced educational approaches including all aspects involved in these phenomena [9,10,11,12,13,14].

## 3. Strategies to Improve Education in Radiation Oncology

About 66% of all cancer patients will receive curative or palliative care during the course of the disease. However, Radiation Oncology is not included in curricula or in many cases, is included as an optional module in the clinical rotation period during university medical studies. De la Peña et al. analyzed the influence of a radiation oncology module, which was presented by radiation oncologist clinicians in the 3rd year of university studies, on the choice of a career in this medical specialization. The study included a test prior to the lectures to assess students’ existing knowledge before any didactical intervention. The proposed lectures included general knowledge of radiation therapy, radiobiology, radiation physics and breast cancer. Ninety-five students were included and the final evaluation showed an improvement of knowledge for all proposed subjects except for medical physics. Based on these data, the authors conclude that it is feasible to introduce a radiotherapy module into the curriculum of the 3rd year of university medical studies to improve students’ knowledge and interest in radiation therapy [15].

The same idea is supported by Golden and collaborators in a study that evaluated the effect of a set of three workshops which included an overview of radiation oncology, radiation biology and physics, and practical aspects of CT simulation and emergency radiation. The multicenter study conducted by twenty-two academic institutions demonstrated a significant improvement in the knowledge of students included in the study coordinated by the Radiation Oncology Educational Collaborative Study Group (ROECSG). Furthermore, the student clerkship in the fourth year of medical studies demonstrated improved knowledge in radiation oncology. All the medical students included in the evaluated educational programs expressed a strong interest in a future career in radiation oncology. The authors support the educational concepts based on structured teaching in radiation oncology clerkships, arguing that the addition of educational clinical programs in medical curricula will also increase the level of knowledge of future resident doctors. Significant gaps in the knowledge of radiotherapy among medical students is a problem reported by many authors. Neppala proposed an interactive contouring module in order to promote the discipline of radiotherapy and provide scientific knowledge in the field to study participants. The analysis of the educational intervention compared in successive evaluations (immediately after the study, at 3 months and later) the effect of an interactive contouring module with the classic lecture method of the radiotherapy course. Even if the differences were not significant in terms of the level of general knowledge in the field of radiotherapy, there was a benefit of +8.6% in the group for which learning by contouring was conducted and a benefit of +6.6% in the group for which the traditional teaching was the preferred method. At 3 months, the differences were no longer significant, but it is worth noting the higher rate of interest in participating in clinical rotation in radiation oncology for the group which used contouring as a learning method. Furthermore, in the group that participated in the learning by contouring of clinical cases, the levels of awareness about the radiotherapy process and the adverse effects of radiation treatment were higher. The authors, therefore, support the introduction of an interactive module of oncological radiotherapy in the preclinical curriculum of medical schools [16,17,18].

The effects of the absence of radiotherapy in the curricula of medical schools worldwide on the career choice of future young doctors is also noted by Arenas et al. Analyzing 25 articles in which the role of educational intervention is evaluated, the authors note that most interventions in the education of medical students belong to the clinical rotation cycle being performed in North America. Most studies used traditional teaching methods, but the interactive method was used in addition to traditional methods. The evaluation methods based on questionnaires addressed to medical students were, however, subjective. Among the frequently encountered topics, the authors note the multidisciplinary approach in oncological disease and psychosocial oncology. In this context, the absence of mentoring programs regarding oncological radiotherapy and research is noted. The introduction of a preclinical oncology course, as well as a radiotherapy session, aimed to evaluate the impact of radiotherapy education on the knowledge gained by medical students regarding the basic concepts in oncology and radiotherapy. Evaluating a group of 319 students who completed the program and also completed the initial and final evaluation the study, Agarwal et al. reported a 7% improvement in knowledge after a didactical intervention. A smaller improvement in knowledge was found in radiation oncology (only 4.4%) compared with the increased knowledge of general, breast and prostate oncology. Although new diagnosed cancers are expected to double by 2040 and oncology teaching initiatives have been already implemented, many students continue to report uncertainties when dealing with cancer patients. A total of 115 papers published in 26 different countries and regions were reviewed to evaluate the optimal method of teaching radiation oncology. Lectures and small group discussions benefited in 97.1% of cases, as well as clinical case simulations, but early mentoring, summer workshops and teamwork benefited in 100% of cases. A knowledge assessment of medical students in Germany based on the response to questionnaires sent to the departments of radiation oncology showed a low level of training in radiobiology and medical physics, of only 25% and 33.3%, respectively. Although most university centers offered resident physicians a rotation in the imaging and nuclear medicine departments, only 70.8% offered students a complete clinical rotation in radiation oncology. In order to improve the educational curriculum, the role of certified courses organized by the German Society of Radiation Oncology (DEGRO) in the training of resident physicians in Germany was evaluated by Büttner et al. using questionnaires. The authors noted an increase from 2018 to 2019 of curricular coverage from 57.6% to 77.5%, but they still considered that this curriculum was not representative and required attention. The authors also mention the need for early integration of education in radiation oncology and suggest that this would have a significant impact on young doctors training in this field. Although traditional seminars have a well-defined role in terms of medical education, the current context created by the new coronavirus pandemic draws attention to the potential of e-learning methods. In Germany, these modern methods are proposed in association with traditional lectures to update the dedicated curriculum for medical students with new data on radiation oncology. The lack of information on current practices in teaching radiotherapy among medical students in Canada is highlighted by Clayton and colleagues who note that one in five students did not receive any information about radiotherapy during their undergraduate medical studies and 65% of the students received less than 2 h of lectures related to the field of radiation oncology. The decision to choose radiotherapy as a specialty in their medical career was more common among students who participated in a medical school program with more than 2 h of lectures. The authors mention the need to introduce the concepts of radiotherapy in the university curriculum for medical students, noting the potential benefit brought in choosing their future medical career [19,20,21,22,23,24,25].

The study of Odiase and collaborators aims to assess the impact and experience of radiation oncology education in medical schools in the United States, including both in allopathic (MD) programs and osteopathic programs. It is noteworthy that although all medical school programs have general oncology curricula, only 4% of students were exposed to radiation oncology specific lectures in years 1 and 2. The study included 198 medical schools and also found a low representation of 42.4% of radiation oncology departments among residents. Only 10.6% of medical schools had an interest group focused on radiation oncology, but 45% of schools had an interest group for general oncology. The authors note less interest in careers in radiation oncology in medical schools and osteopathic medical schools where there is no mentor in this field. At least one student match in radiation oncology is a scenario associated with older medical schools with mentors. The authors note the correlation between the lack of exposure to radiation oncology lectures and students’ lower interest in the field, mentioning the need for their study to be considered both nationally and in medical schools. Arifin’s study reviews 25 articles that include topics related to radiation oncology education, finding that in North America a single session of teaching for medical students is the most widely used method. The authors note that multidisciplinary oncology and psychosocial oncology are common topics, with didactic teaching followed by interactive teaching being the preferred methods. The study also notes the paucity of research and formal mentorship topics [26,27].

The concept of mentorship was defined by Healy and Welchert as a dynamic and reciprocal relationship between the career mentor and a beginner and is considered a beneficial relationship for both. Even though it has elements in common with activities such as teaching, leadership and advisory roles, it is important to mention that mentoring is a more complex concept. Traditionally, mentorship is based on the dyad consisting of a single mentor and a single junior mentee. Job retention, mentee satisfaction and promotion are already recognized advantages that benefit the mentee, but the mentor also gains skills, satisfaction and camaraderie in the team. The study by Marsiglio and collaborators aims to assess the status of mentorship in radiation oncology to promote and improve these initiatives. The authors examined eight types of mentorships: dyad, multiple dyad, functional dyad, speed mentoring, distance mentoring, team mentorship, peer mentorship, and facilitated peer mentorship. Thirteen publications from 2008 to 2019 considered relevant to the topic were included in the study. It should be noted that although the study searched for articles published after 1990, no papers were published on this topic until 2008. The study methods included surveys, productivity metrics and semi-structured interviews. Six of the studies reported satisfaction of mentees, mentioning values as “satisfied” or “very satisfied”. Formal mentoring activity has also been associated with a higher levels of satisfaction than informal mentoring. Reasons for dissatisfaction such as difficulty finding a mentor, inexperienced mentors, disinterest of the mentor and a lack of time for mentoring were also mentioned in the study [28,29].

## 4. Radiation Oncology Education—The Offer of Professional Organizations

Without intending to include all the educational efforts proposed by the national and international scientific societies, Table 1 summarizes some of the directions proposed by the main radiation oncology professional and scientific societies as well as the International Atomic Energy Society Agency. We also mention the limited access to information about educational approaches in the official websites of national societies, due to the non-existence of an international language version these websites [23,30,31,32,33,34,35,36,37,38,39,40,41,42,43].

## 5. Conclusions

Education in radiation oncology is a major priority in the context of the rapid development of radiotherapy, including advanced knowledge of radiobiology, radiation physics and clinical oncology, anatomy, tumor biology, and advanced medical imaging. In this context, the lack of training in radiation oncology in the curricula of medical faculties can have detrimental consequences for the training of residents in radiotherapy and for the choice of specialty after completing university studies. The introduction of radiation oncology in the medical education curriculum, both in the cycle of preclinical studies with basic concepts in radiotherapy, medical physics, radiobiology and also in the period of clinical rotation, can bring long-term benefits in increasing the level of training for resident physicians and interest in radiation therapy. For resident physicians training, considering the possible low exposure of medical students to radiotherapy and considering the difficulties and challenges of education in such a complex interdisciplinary field, only radiation oncologists involved in educational activity for medical students should coordinate this teaching activity. The gap between the actual and required levels of knowledge among resident physicians about radiotherapy requires urgent remediation. In the context of technical advances in imaging-guided radiotherapy (IGRT) and new radiobiology data, a balanced approach divided equally between general oncology, clinical radiation oncology, radiation oncology technology, medical physics and radiobiology, anatomy and multimodal imaging could bring substantial benefits to education in radiation oncology.

## Figures and Tables

**Table 1 ijerph-19-03772-t001:** Educational initiatives of international and national radiation oncology societies [23,30,31,32,33,34,35,36,37,38,39,40,41,42,43].

RO Education Area	Proposing Organization (Radiation Oncology Society)	Educational Programs
Biostatistics/Clinical trials	ASTRO SRO	patient reported outcomes (PROs), predicton of patient outcome courses, patient-reported outcomes of oligometastatic patients after conventional or stereotactic radiotherapy to bone metastases, one radiotherapy and immunotherapy clinical trials design course medical statistics and clinical trial design
Radiobiology	ESTRO ASTRO SFRO	clinical radiobiology course biomarkers in radiotherapy, radiotherapy and in-situ vaccination, liquid biopsies in radiation oncology eLQ—iOS application, radiobiology course
Contourig/Treatment planning	ESTRO ASTRO SFRO SRO RATRO	11 site specific workshops including liver SBRT and OARs/9 treatment planning courses with MRI-guided RT, IMRT/VMAT, SBRT, adaptive radiotherapy 3 SBRT courses including lung cancer and oligometastasis siriade online application, radioanatomy and brachytherapy courses modern techniques in radiation oncology contouring workshops
Site specific cancer	ASTRO SFR BeSTRO SEOR	12 site specific courses 12 site specific courses including rare tumors topics 5 site specifc online courses masterclass of non-small cell lung cancer question based content about prostate cancer
Research	ESTRO SFRO	radiotherapy physics, risk management and patient safety courses, radio-physics course
Physics	ESTRO ASTRO	physics workshop cardiac radio-ablation for cardiac arrhythmias, introductory physics modules for radiation oncologist residents
Other	ESTRO ASTRO SFRO SFO AIRO SEOR DEGRO CARO BeSTRO RCR	moderns skills, leadership, radiation therapist training courses, palliative care, geriatric oncology, department response to cyber-attack, geographic access to radiation therapy facilities in the United States, health policy (federal and private health insurance payers), Best of ASTRO sessions, 3 courses about radiotherapy and immunotherapy in clinical practice, RO-ILS: Radiation Oncology Incident Learning System^®^, patient safety and error detection in radiation oncology courses summer schools, how to talk to patients, mock Exam (iPad), exam information, psychooncology, mentoring program palliative and rehabilitative oncology Joint project for education of radiation therapists and medical physicists in interventional radiotherapy (brachytherapy) university forum for undergraduate cancer education, radiation oncology teaching for medical student initiatives DEGRO-Academy (German curriculum-based residency training programs in radiation oncology) leadership in radiation oncology course re-irradiation and systemic therapy online courses leadership in radiation oncology online course

ESTRO—European Society of Radiation Oncology; ASTRO—American Society of Radiation Oncology; SRO—Swiss Society for Radiation Oncology; SFRO—French Society For Radiation Oncology; AIRO—Italian Association of Oncological Radiotherapy; DEGRO—German Society of Radiation Oncology; RATRO—Russian Association of Therapeutic Radiation Oncologists; RCR—The Royal College of Radiologists; SEOR—Spanish Society for Radiation Oncology; CARO—Canadian Association of Radiation Oncology; BeSTRO—Belgian Society of RadioTherapy Oncology.

## Data Availability

Not applicable.

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
