# Peer review of "Education in Radiation Oncology—Current Challenges and Difficulties"

_ijerph, 2022, doi:10.3390/ijerph19073772_

Round 1
Reviewer 1 Report
This appears to be restricted to some country / region ,
however the problem is global , may see some distant learning methods from IAEA , self learning modules etc from different professional societies which can be included. The authors should provide the list of resources in some form of table for each important area ; i.e , Biostatistics , radiobiology and so on.
Author Response
Dear Reviewer,
Thank you very much for your time in evaluating this manuscript and for your suggestions. We introduced a table which summarized educational initiatives of the main international and national scientific societies, We introduced a paragraph with the educational programs proposed by the IAEA. We have also introduced a short paragraph in which we mention some of the "new" toxicities brought about by advances in multimodal cancer management and the need for a deeper understanding of all the phenomena involved. We corrected the unclear sentences and corrected the reported mistakes. We have also introduced new references, especially official websites of international and national professional radiotherapy companies.
We hope you the proposed changes and look forward to hearing from you new suggestions to improve the article.
Kind Regards,
Camil Mirestean
Reviewer 2 Report
The manuscript is of interest and addresses the relevant topic of education in radiation oncology.
I would suggest the authors to add a brief paragraph on the difficulties and complications arising following radiation therapy, in order to clarify why the issue of radiation oncology education is relevant.
The section entitled "Strategies to improve education in radiation oncology" should provide a starting point for a Discussion section, which is lacking in the present version of the manuscript. The authors are encouraged to critically analyze the evidence from the previous literature, and to discuss the more effective approaches in providing adequate education.
Author Response
Dear Reviewer,
Thank you very much for your time in evaluating this manuscript and for your suggestions. We introduced a table which summarized educational initiatives of the main international and national scientific societies, We introduced a paragraph with the educational programs proposed by the IAEA. We have also introduced a short paragraph in which we mention some of the "new" toxicities brought about by advances in multimodal cancer management and the need for a deeper understanding of all the phenomena involved. We corrected the unclear sentences and corrected the reported mistakes. We have also introduced new references, especially official websites of international and national professional radiotherapy societies.
We hope you the proposed changes and look forward to hearing from you new suggestions to improve the article.
Kind Regards,
Camil Mirestean
Reviewer 3 Report
The manuscript discusses the importance of education in radiation oncology and of training of resident physicians in radiotherapy. This could help in guiding the choice of specialty towards this innovative and rapidly evolving field of medicine. Possible strategies in bridging this gap in the curriculum of medical schools are also suggested. The topic is of clear interest in the medical field and it is sufficiently addressed and discussed by the authors. I recommend publication upon revision of minor typos and a few unclear sentences.
Specific comments:
- Line 12, 29: 2 decades, change with two decades
- Line 24-25: this sentence is not clear
- Line 51: change server with severe
- Line 110: remove to
- Lines 127-129: this sentence is not clear
- Line 134, 143, 337: change rotatio with rotation
- Line 245, remove is
Author Response

(The authors gave the same response as above.)

Reviewer 4 Report
Camil Ciprian Mirestean and colleague have presented their opinion titled “Education in Radiation Oncology - Current challenges and difficulties”
In this study authors have highlighted the problem of the gap between the level of knowledge of radiotherapy and the need for knowledge of a resident physician.
Authors have emphasized on the need of proper education in radiation oncology is a major priority in the context of the rapid development of radiotherapy, including advanced knowledge of both radiobiology, radiation physics and clinical oncology, anatomy, tumor biology and advanced medical imaging. In this context, the lack of training in radiation oncology from the curriculum of medical faculties can have detrimental consequences in the training of residents in radiotherapy but also in the choice of specialty after completing university studies.
I think the study is nicely articulated and presented. This study will be useful for the researchers and clinician working in this field.
Therefore, I recommend accepting it.
Author Response

(The authors gave the same response as above.)
